# The Misguided Veneration of Averageness in Clinical Neuroscience: A Call to Value Diversity over Typicality

**DOI:** 10.3390/brainsci13060860

**Published:** 2023-05-26

**Authors:** Graham Pluck

**Affiliations:** Clinical Cognitive Sciences Laboratory, Faculty of Psychology, Chulalongkorn University, Borommaratchachonnani Srisattaphat Building, 254 Phayathai Road, Bangkok 10330, Thailand; graham.ch@chula.ac.th

**Keywords:** neurodiversity, cognitive assessment, neuropsychological assessment, normative data, central tendency, evolution, degeneracy, premorbid function, idiographic, psychometrics

## Abstract

Research and practice in clinical neurosciences often involve cognitive assessment. However, this has traditionally used a nomothetic approach, comparing the performance of patients to normative samples. This method of defining abnormality places the average test performance of neurologically healthy individuals at its center. However, evidence suggests that neurological ‘abnormalities’ are very common, as is the diversity of cognitive abilities. The veneration of central tendency in cognitive assessment, i.e., equating typicality with healthy or ideal, is, I argue, misguided on neurodiversity, bio-evolutionary, and cognitive neuroscientific grounds. Furthermore, the use of average performance as an anchor point for normal performance is unreliable in practice and frequently leads to the mischaracterization of cognitive impairments. Examples are explored of how individuals who are already vulnerable for socioeconomic reasons can easily be over-pathologized. At a practical level, by valuing diversity rather than typicality, cognitive assessments can become more idiographic and focused on change at the level of the individual. The use of existing methods that approach cognitive assessment ideographically is briefly discussed, including premorbid estimation methods and informant reports. Moving the focus away from averageness to valuing diversity for both clinical cognitive assessments and inclusion of diverse groups in research is, I argue, a more just and effective way forward for clinical neurosciences.

## 1. Introduction

Historically, clinical neurosciences, such as neuropsychology, neuropsychiatry, and behavioral neurology, have tended to use a nomothetic approach, focusing on how general principles common across all people explain behavior in health and disease. Although fields such as neuropsychology and behavioral neurology have classically used the lesion–symptom association approach, identifying cases with unusual abilities due to neurological illness, the comparison and inferences drawn have generally been to the ‘normal’ state of psychological ability [1]. A comparable pattern is seen in fields such as neuropsychiatry, which also often interprets signs and symptoms in cases in terms of the framework of healthy functioning [2]. Similar points can be made for other fields that deal with neurodevelopmental disorders. Thus, clinical symptoms and signs have been predominantly interpreted as a deviation from the common pattern seen in the neurologically healthy. Here, I argue that the comparison of the cognitive performance of an individual to what is typical or average is misguided for various biological, cognitive, and psychometric reasons.

## 2. Neurological and Cognitive Diversity

In recent years, diversity, as opposed to commonality, has increasingly been recognized among cognitive neuroscientists. This has led to a more idiographic approach interested in the idiosyncratic nature of human experience, cognition, and brain functioning. One reason for this is that it is now clear that being ‘neurologically healthy’ is not as common as might be assumed. One study of over 1000 individuals aged 50–66, selected from the general population, found that 22% had one or more incidental abnormalities on brain MRI scans, and a further 5% had previously recognized findings [3]. Several other neuroradiological studies in healthy samples have reported incidental findings on MRI scans, though with substantial variation in the observed prevalence of observation. A recent review of these suggests that in non-clinical adult samples, the prevalence of incidental neurological findings ranges between 9 and 54% [4]. Thus, some levels of neurological ‘abnormality’ is relatively normal.

In parallel, normal variation in the phenomenology of cognition has become increasingly recognized. Of course, it is well known that individuals vary in their proficiency in many cognitive tasks, with a more-or-less linear spectrum of performance. However, it has only relatively recently been noted that individuals may also vary in the type of cognitive processes that they can perform, suggesting a non-linear discontinuity akin to the presence or absence of disorder. Although most people experience visual imagery, some people do not and have not had such experiences for all of their lives, which is a condition named congenital aphantasia [5]. It has an estimated prevalence in the general population of about 1%, with a further 3% reporting very limited mental imagery [6]. Similarly, prosopagnosia, originally considered an acquired neurocognitive disorder, has been shown to occur in a minority of people connaturally [7], with a prevalence of about 2% [8]. A related interindividual variation is an ability, or not, to recognize voices, called developmental phonagnosia, with a prevalence of about 3% [9]. In a similar vein, synesthesia, the atypical association of sensory stimuli and perceptual experiences, is a known consequence of brain injury [10] but may occur naturally in up to 4% of people without any known neurological disorder [11].

Such discoveries of the presence, in healthy individuals, of psychological phenomenon hitherto considered to be consequences only of disease of the nervous system have led some clinical neuroscientists to a greater appreciation of normal variation in types of cognitive processes available to individuals. This enlightened recognition is timely, as it coincides with a grassroots movement to value diversity associated with what have previously been considered neurodevelopmental ‘disorders’. This movement, known as neurodiversity [12,13,14,15], views variations in brain function, such as those associated with the autism spectrum and attention deficit hyperactivity disorder, as normal rather than disease states. These variations produce behavior that may be different from that of most people but are part of normal human variation. Neurodiversity is, thus, defined as the ‘normal range of function in a population’ and is ‘a characteristic of the whole population, not a specific individual’ [14]. Despite being originally driven by user groups, its broad perspective of emphasis on diversity, not disorder, has been rapidly accepted by eminent developmental neuroscientists, e.g., [13,15]. Recognition of neurodiversity can lead to more appropriate adaptions and perhaps interventions for behaviors that are impinging on daily activities [16].

However, this progress in recognizing the normality of neurodiversity contrasts with a strong tradition, particularly in cognitive assessment, that explicitly values typicality. This is the tendency to use statistical central tendency to formalize averageness and, consequently, define what is normal and what is not normal based on deviation from the average. This veneration of central tendency in many aspects of clinical neuroscience is expressed most obviously in cognitive assessments used to identify clinical impairments. This is because in almost all forms of cognitive assessment, normative data, anchored around the central tendency of a test sample, is used to demarcate impaired and nonimpaired performance.

This core belief in central tendency (i.e., averageness) as an ideal is the antithesis of valuing diversity. However, it is not only a misassumption in terms of social justice but is both unjustifiable from bio-evolutionary and neurocognitive perspectives and is unreliable and limited in practice.

## 3. The Use of Central Tendency in Cognitive Assessment

Clinicians who perform cognitive assessments need to identify which abilities have been impaired by neurological or psychiatric illness and evaluate the extent of the impairments. Even in research, the majority of published neuropsychology journal articles are on the subject of describing impairments associated with neurological or psychiatric illness or developing new tests to do so [17]. The problem faced in both research and clinical practice is that those attempting to detect cognitive impairments rarely ever have the opportunity to measure abilities before the onset of the illness, making absolute measurement of change impossible [18]. Instead, they must use clinical judgment to best estimate psychological changes consequent to the illness impinging on brain function.

In the most developed countries, where cognitive test development is most prevalent, i.e., those described as WEIRD—Western, Educated, Industrialized, Rich, and Democratic [19], this is typically achieved via data tables derived from the task performance of representative samples of individuals. Some central tendency point in the score distribution is identified, usually the mean. Then, a prespecified range around that anchor point is identified as normal performance, and anything falling outside that range is identified as abnormal. The normal range is usually defined in standard deviations from the mean of the data distribution. Sometimes the central tendency point is the 50th percentile (the median), but the system of defining normality and abnormality based on central tendency is the same. In such systems, the normality or abnormality of test performance is fully defined by the statistical properties of the representative sample used to generate the normative data tables. This can be seen as an inter-individual method of defining abnormality. The most common example of this is the use of IQ scores (e.g., with the Wechsler tests), which are derived from a normative distribution that is transformed to have a mean of 100 and a standard deviation of 15. Abnormal performance consistent with mental retardation is defined as an IQ score more than two standards below the mean, i.e., 69 or below [20]. However, the same principles are applied to common assessments of amnesia, aphasia, attentional disorders, etc.

A major concern with this approach is that, although the central tendency is used to anchor the range of normality, the borders between that range and what will be identified as abnormal performance are arbitrary. Some systems delineating normal from abnormal performance use two standard deviations from the central tendency point, equivalent to the 2nd percentile; others use the 5th or 10th percentile, while others have used one standard deviation, equivalent to the 16th percentile [21]. Even the 40th percentile has been noted [22], which would classify a very large proportion of healthy individuals (40%) as cognitively impaired.

Furthermore, as cognitive assessment typically uses multiple tools, the risk of miscategorization is repeated several times. Binder et al. [23] found that when applying 20 or more cognitive tests, if one standard deviation below the mean is taken as indicating abnormality, then the majority of healthy individuals would have at least one abnormal score. This led the researchers to suggest that, paradoxically, “abnormal performance on some proportion of neuropsychological tests in a battery is psychometrically normal” (p. 1). The implication of these effects is that the ostensibly dispassionate use of statistics to identify normality via deviation from the central tendency often fails to disambiguate abnormality from normal diversity.

This task is further complicated by the substantial overlap between normal cognitive performance (for some people) and impaired performance (for others). This can be revealed by comparing the inter-individual method described above with an intra-individual method in which a drop in scores from pre- to post-illness is calculated at the level of the individual but still represented in the same metrics used with normative tables. Take, as an example, a patient who has suffered a drop in fluid intelligence test performance of one standard deviation (of the estimated population distribution); that one standard deviation magnitude of intra-individual change would be considered consistent with some definitions of significant decline or diminished cognitive ability [24]. However, the normal inter-individual range of scores is commonly described as spanning four standard deviations, which would be 70–130 on a Wechsler scale [20]. On that criterion, the magnitude of a diagnosable cognitive decline is only one-fourth of the normal range in abilities. Furthermore, an individual could suffer a significant clinical decline while remaining above average in their ability.

Clinical neuroscientists involved with cognitive assessments have often neglected to account for this overlap and relied on the central tendency within samples as a representation of normal ability (i.e., using an inter-individual approach). Thus, the central tendency point from a representative sample has been taken as the point of typical performance. These central tendency anchor points are explicitly used in clinical neuropsychology and behavioral neurology to estimate the original (premorbid) performance of an individual with neurological illness [25,26,27]. Even if the use of the central tendency of a normative sample is not explicitly stated as being used to estimate premorbid ability, the fact is that abnormality is routinely identified as being any ability that is sufficiently different from that average point.

Measures of central tendency, such as the mean or 50th percentile, are defined as ‘the statistical measure that identifies a single value as representative of an entire distribution’ [28]. In this sense, the mean is used to provide a ‘portrait of a typical participant’ [29] or the performance of the ‘ultimate representative subject’ [30]. The logic is then that if the currently measured performance of an individual falls substantially below that typicality point, it can be considered ‘abnormal’ [18]. This emphasis on the performance of the ‘ultimate representative subject’ explicitly values typicality and devalues diversity. In fact, the devaluation of non-typicality is applied inconsistently, as the emphasis is almost always on what has been predefined as low scores, despite high and low scores being equally and symmetrically ‘abnormal’.

## 4. The Biological–Evolutionary Perspective: Diversity of Abilities and Traits Is Normal

In the pre-evolutionary period, stemming from Greek philosophy, biological science was focused on ideals in which species were defined by their commonality, ignoring individual characteristics [31]. When clinicians perform cognitive assessments based on the central tendency, they continue to make this error, assuming that there is an ideal point of healthy neurocognitive functioning. At its most basic level of biological analysis, we may consider whether better neurocognitive functioning is fundamentally better.

It is true that larger brains are associated with better cognitive processing [32]. However, it has often been erroneously thought that bigger and better brains, and consequently, better cognitive processing, are a goal of human evolution, e.g., [33]. The problem with that assumption is that brains are metabolically expensive organs to possess. In humans, despite only being about 2% of body mass, the brain consumes about 21% of available energy [34,35]. This is a substantial cost to the individual, amongst other factors, caused by the energetic cost of continuous ion pumping and manufacture of neurotransmitter substances, and is particularly acute during learning which places high demands on energy resources [34,35,36]. Thus, although there is always evolutionary pressure for better adaptive behavior, there is simultaneous and continuous evolutionary pressure for reductions in brain size [36]. This pressure is bidirectional as environments vary and the relative need for costly but adaptive neurocognitive processing varies, such that in some ecological circumstances, brains will evolve with reduced size and cognitive processing capacity [37]. In fact, the human brain has been subjected to these forces, and the average human brain size has been getting gradually smaller over at least the past three thousand years, likely due to societal development resulting in reduced demand for active cognitive processing by individuals [38].

Consequently, neither higher nor lower intelligence is fundamentally better. The point is that from a purely biological perspective, for an individual to thrive, there is no absolute optimum level of neurocognitive functioning. Where optimum levels exist, these will be relative to the environments that people live in.

Even within specific environments or niches, a diversity of cognitive ability is completely normal. The use of central tendency points in normative data erroneously attempts to anchor a range of cognitive processing abilities that is typical and, from that, ranges that are said to be atypical. Although a central tendency point can easily be found in a data set, this does not mean that an optimal point has been located, one that can be considered ‘normal’ across different samples. On the contrary, inter-individual variability in functioning is completely normal. This is a further consequence of the processes of evolution in general, including the evolution of the brain. Within a species, such as humans, variability within traits is crucial for the species’ survival. This diversity in traits is an essential component of evolution; without it, there could be no natural selection [39].

An example from human individual differences research is the Big-5 personality traits, which represent variation in personality across populations and are substantially heritable [40]. People phenotypically vary in the extent to which they manifest the identified traits of Openness, Conscientiousness, Extraversion, Agreeableness, and Neuroticism. This variation exists rather than evolving into optimal set points of ‘normal’ personality because environments vary, and each heritable personality phenotype has costs and benefits dependent on those varied contexts [41]. As a timely example, individuals who have neurotic personalities are, in general, at a substantially increased risk of developing affective disorders [42]. However, early in the recent coronavirus (COVID-19) pandemic, the extremely cautious attitude of neurotic individuals meant that they were less likely to become infected by the virus [43], providing a survival advantage.

Of course, this psychological diversity, which necessarily involves costs and benefits, is not limited to personality. Diversity in cognitive abilities is normal, at least for those aspects which are heritable [44]. Biological and cognitive phenotypes can be adaptive or not, depending on the context that the individual lives in and even their age. From a biological perspective, individuals that have the ε4 allele of the apolipoprotein E gene (APOE4) are at substantially increased risks of developing Alzheimer’s disease, Lewy body disease, cerebrovascular disease, and accelerated age-related cognitive decline [45,46]. The phenotypic variation appears to be expressed early in life, as newborn children that carry the APOE4 variant have reduced volumes of the hippocampus and other temporal regions, as well as larger volumes in the parietal lobes, compared to carriers of other variants [47]. Furthermore, healthy carriers of APOE4 also show subtle cognitive impairments in mid-adulthood, affecting attentional and working memory processes, suggesting that even in the absence of dementia, the phenotype includes cognitive impairments [48]. This seems to indicate a genetically mediated individual difference biomarker for neurocognitive dysfunction.

In contrast, possession of the APOE4 variant may have positive effects on young people. In a sample of women aged 19–21, possession of the APOE4 variant was found to be associated with higher performance IQ [49]. Carriers are also more likely to complete higher education [50]. One of the mechanisms for this, at least in low- and middle-income countries, is that carriers of the APOE4 variant have substantial protection against childhood diarrhea and parasitic infections, which may be why it remains a common variant within the gene pool [51]. In fact, amongst children living in poverty, there is a higher than the would-be-expected number of individuals positive for the APOE4 variant, likely because of its contribution to survival via the prevention of enteric diseases [52]. In addition, diarrhea and parasitic infections during the first two years of life are important predictors of cognitive ability in later childhood [53]. Thus, possession of the APOE4 gene variant may predispose carriers to dementia in later life but may enhance cognitive development early in life due to better intestinal functioning and nutrition [54], particularly for individuals living in environments with high pathogen exposure. This demonstrates the point that specific variants in neurocognitive traits can be beneficial or harmful in different contexts.

Much of the research that revealed this association between better cognitive development, intestinal disease, and APOE4 involved infants and children living in Brazilian shanty towns [52,54,55], individuals usually neglected in clinical neuroscience research. Their inclusion in studies has undoubtedly increased understanding of biomarkers for cognitive function and cognitive disorder, in this case, demonstrating that individual differences, such as APOE phenotypes, may have survival and development benefits. A wider diversity of research participants in clinical studies is likely to reveal another phenomenon that might be missed in research restricted to WEIRD populations. This will help neuroscience to see around its ‘diversity blind spot’ which has demonstrably led to several incorrect theories about brain function, such as the erroneous conclusions about hemispheric specialization that resulted from fMRI studies that excluded left-handed participants [56].

Related to this, the belief in the superiority of delayed gratification over immediate reward, is common in psychology and neuroscience research, e.g., [57], but ignores the fact that advantageous action in resource-limited environments, such as poverty, may be different from those where resources are less limited. This bias partly comes down to the widespread but erroneous belief in a primitive ‘emotional brain’ that competes with a more developed and evolutionary recent ‘rational brain’ [58].

Researchers who have investigated more diverse populations, including those living in poverty, have often revealed surprising results that do not fit with the idea of continuums of ability with optimum performance points. Homeless street children in Bolivia achieve higher scores on divergent thinking than never-homeless children [59]. Children working in carpet factories in Nepal achieve higher scores on working memory compared to school-attending peers [60]. Children working in Brazil as street vendors score higher on arithmetic tests than school-attending children [61]. Compared to higher social-class children, lower social-class children excel in a range of social cognition tasks [62]. Although these advantages are likely developed through experience, rather than being natural traits, what is ‘normal’ cognitive performance clearly varies across diverse life experiences, which again does not fit with the idea of the average ‘typical participant’ [29], or the performance of the ‘ultimate representative subject’ [30], concepts that are core to the use of central tendency and normative tables for the interpretation of cognitive assessments.

Further examples of the importance of phenotypic diversity over statistical normality come from the relationships between neurocognitive variation and vocation. Some phenomenological states, hitherto identified as pathological, may actually be adaptive. Synesthesia appears to be more common in art students [63], and aphantasia may be over-represented in scientists [64]. These associations suggest that the expression of these extremes of diversity may confer an advantage, depending on occupational context. Similarly, high expression of traits on the autistic spectrum is associated with scientists and mathematicians [65], while expression of traits linked to psychosis is raised in professional comedians [66]. While a naïve biomedical interpretation would anticipate that such traits are pathological and past some threshold, indicative of impairment, they may, in fact, confer a professional advantage, dependent on context. These observations are consistent with the previously mentioned concept of neurodiversity, which argues that neurodiverse individuals, compared to neurotypical individuals, have both advantages and disadvantages, depending on the context [13,14,15]. In particular, the adaptiveness of neurodiversity in occupational roles has been examined in detail [12]. Others who favor the neurodiversity approach have suggested an ecological model that similarly examines the role that diversity of abilities has in adaptability, particularly for ensuring future success in changing environments [67].

Overall, the evidence from a bio-evolutionary perspective shows that the diversity of neurocognitive traits is normal, that such traits have costs and benefits that vary by context, and that there is no such thing as an optimal brain function profile [68]. This subtlety is missed when the abilities of individuals are compared to some predefined normal/average point.

## 5. The Neurocognitive Perspective: Degeneracy

There are biological problems with using central tendency to represent the typicality of cognitive ability. A key issue is degeneracy. This is the ability of different biological structures to produce the same functional outcome [69,70]. Single-word reading ability provides a useful example of this. Kherif et al. [71], using functional magnetic reasoning imaging (fMRI), found that across a large group of participants who read words aloud, four different patterns of brain activation could be identified. While all participants activated the occipitotemporal sulcus, presumably involving word recognition, and motor cortex, presumably involving articulation, they varied in their activation of other areas, particularly of the dorsal and ventral frontal cortex. Importantly, the different groups had an equivalent performance on a range of verbal ability and reading tasks. Similarly, Seghier and Price [72], also using fMRI, revealed that a key moderator of activation between areas related to word recognition and articulation was activity in the putamen, as some participants used that route between posterior and anterior cortices, while others did not.

This indicates that some neurocognitive functions show degeneracy. In fact, such degeneracy would be predicted from cognitive and neuropsychological models of word reading, which have proposed multiple routes from word reading to pronunciation, e.g., [73]. A particularly well-known example of this is in pure alexia, in which brain-injured patients lose the ability to recognize whole words but can compensate to almost normal levels of word-reading ability through letter-by-letter reading, with sufficient time to develop that alternative strategy [74]. Obviously, this compensatory strategy could make reading ability appear normal if it were compared to normative data that was based on healthy individuals who were able to use the whole-word-reading route. Thus, true impairments could be missed through the degeneracy of neurocognitive processes in reading.

In fact, many other cognitive and neurophysiological processes show degeneracy. Blindsight, the ability to act on visual information despite the subjective experience of blindness following damage to the primary visual cortex, can be viewed as one example. Blindsight clearly shows that alternative neural routes can be used to make decisions based on input from the eyes, such as via the superior colliculus or pulvinar [75]. Similarly, action imitation, a skill commonly assessed by neurologists, may be produced by separate neurocognitive routes [76]. Degeneracy appears to be a common feature of normal brain physiology, working not only at the system’s level but also down to the cellular, molecular, and genetic levels [70].

In a sense, such degeneracy of functions should not be surprising; it is obvious that research participants and clinical patients vary in their approach to task performance, for example, by applying strategies. However, focusing on multiple methods and neural routes reveals two main problems with comparing individuals to some idealized point based on aggregate statistics. Firstly, if there are multiple possible available systems to perform some cognitive tasks, then some lesions that nevertheless impair relevant neurocognitive processes will not produce any impairment in task performance. Other lesions will. When data is pooled over patients to create averages of performance, incorrect lesion-symptom associations can be made [77]. More importantly for the current analysis, the aggregation of scores from samples of healthy individuals may, in fact, be merging very different processes, producing statistical artifacts. Following this, attempts to compare individuals to the central tendency of a normative sample will ultimately be misleading and risk misidentifying normal performance as abnormal and vice versa.

This problem with averaging across individuals who may be using very different neurocognitive strategies has been previously noted in other areas of brain research, including behavioral neuroscience [78] and functional neuroimaging [30]. Within neuropsychology, cognitive control processes may be particularly susceptible to this, as they often involve strategy. Simon [79] has, for example, identified at least four different ways that people can complete the Towers of Hanoi task (a commonly used clinical assessment of executive function), with varying dependence on short-term memory, perception, and learning mechanisms. Given such diversity of processing types, it is unclear what exactly an individual’s performance is being compared to when it is referenced to the average from a sample (i.e., normative data).

There are other issues concerning the dynamism of cognitive processing and the use of averages of performance. One of these is the degree to which cognitive processes may be modular or functionally dependent. Although that debate is beyond the scope of this paper, it should be noted that some impaired performance following neural insult can be explained by non-specific reductions in general intelligence. Comparing patient performance on some cognitive tasks to average performance from a normative sample may thus give the impression of focal impairments when none exist. Again, cognitive control processes may be particularly susceptible. Roca et al. [80] have shown that following frontal-lobe damage, impaired performance on several common tests of executive function can be fully explained by reductions in general intelligence. Similarly, some neurocognitive processes may act antagonistically, which will not be adequately captured in estimates of the central tendency of task performance [81,82].

## 6. (un)Representativeness of Samples

From a practical perspective, the most common use of averageness to define typical performance in clinical neurosciences is with the use of normative data tables of common cognitive assessment tools. Such tables are typically developed from large groups of individuals thought to be representative of the general population. However, often the representativeness is highly questionable. For example, the normative data for a commonly used memory assessment, the Hopkins Verbal Learning Test—Revised [83], is based on a USA-based sample that contains three times as many women as men. Additionally, a commonly used clinical assessment of executive functioning, the Wisconsin Card Sort Test 64 Card Version [84], uses an adult USA-based normative sample that is partly composed of ‘students and friends of students’ (9%) and ‘commercial airline pilots’ (28%). Thus, when a clinical cognitive assessment is made with that tool, the patient is effectively evaluated as to how well they perform relative to college students and pilots.

Even when recruited samples are stratified by age, sex, etc., to make them seemingly representative of the general population, exclusion criteria that are applied to remove variance in performance for non-cognitive reasons can inadvertently render samples unrepresentative. For example, Weschler intelligence scales, e.g., [85], Wechsler memory scales, e.g., [86], and the Delis–Kaplan Executive Function System [87], three of the most widely used cognitive assessment systems, used more or less the same set of exclusions for their standardization samples. These were collected in the USA and said to be representative and matched to US census data on ‘sex’, ‘race/ethnicity’, and ‘educational level’. However, among the many exclusion criteria, the following were included: visual, hearing, or motor disabilities; non-English speaking, which would have left out about 8% of the US population at the time of the norming [88]; use of medications for mental illness, which would exclude about 17% of the adult US population [89]; exclusion for drinking at least three alcoholic drinks more than twice a week, which would exclude perhaps 8% of the US adult population [90]. Similarly, the exclusion of individuals with color blindness would lead to about 7% of men and 1% of women being excluded [91]. After such exclusions, the remaining sample is, therefore, unlikely to represent the actual population.

There are two important implications of these exclusions when forming normative samples. Firstly, from a social justice perspective, the developers are implicitly valuing typicality at the cost of true human diversity. The foundations of these tests are then biased towards a predefined vision of normality. Secondly, from a practical perspective, any decisions made based on judging typicality against the norms will be misleading.

Furthermore, when those samples were collected, individuals with conditions that could affect cognitive functioning were also excluded. This would have had the effect of truncating the distribution at the lower performance tail, such that the resultant ‘impaired range’ would be composed of scores that would not have been there if a truly representative sample had been used [92]. The child sample of the Wisconsin Card Sort Test [84] is an example of a normative distribution in which the lower ability tail is truncated. This is because children with neurological disease, learning disability, and emotional or attentional disorders were excluded from the sampling. It is also worth noting that, despite such exclusions, and as described above, many people have neurological findings that they are unaware of, perhaps at levels of over a quarter of the population [4]. Consequently, the normative samples inevitably do contain much data from individuals with neurological disorders.

This lack of true representation in ostensibly representative samples of neurologically healthy individuals is not simply an academic point lacking evidence of actual ensuing problems. The Boston Naming Test, a widely used aphasia assessment, is a case in point. Several different normative data tables have been produced in North America, each producing quite different percentile scores for the same test performance. For example, the same task performance could be considered impaired (at the 2nd percentile) or completely average and unimpaired (at the 50th percentile), depending on which normative tables are used [92]. Similarly, when a sample of healthy participants was tested on both the California Verbal Learning Test (CVLT) and the revised version (CVLT-II), they scored, as would be expected, almost identical raw scores. However, individuals achieving ‘normal’ T-scores of 50 (the exact central tendency point) with the CVLT-II could produce ‘abnormal’ scores more than two standard deviations lower when using the older CVLT and associated normative tables. This was because the demographics of normative samples of the CVLT and CVLT-II were so different [92].

## 7. Normative Data from WEIRD People

Representative sampling to produce normative data tables is an expensive, complex, and time-consuming research endeavor. For use in the USA, the Weschler Adult Intelligence Scale IV (WAIS-IV) [20] was normed on 2200 individuals, while the D-KEFS was normed on 1750 [87]. Such large samples likely help to avoid some of the problems described above. However, such normative ventures are major investments and only likely to be profitable in the world’s largest and most economically developed countries. Even then, the market economics, such as how often cognitive evaluations are requested in clinical situations and how much organizations are willing to pay for them, limit the commercial viability of large-scale norming. Indonesia has a large population comparable to that of the USA. However, as a lower–middle-income country, it has limited resources available for neurological care, for example, having only 11 MRI units [93]; the comparable number in the USA is approximately 12,000. Consequently, in Indonesia, resources for large-scale norming of cognitive tests are scarce. In low- and middle-income countries, such as Indonesia, very few cognitive tests have clinical-assessment quality normative data available, including very common tests such as the WAIS-IV. This situation occurs in most countries, affecting most of the world’s population. Norming on cognitive assessment tools is a WEIRD country issue, and even then, tests may only be normed in the most populous and wealthy countries.

In the absence of country-appropriate normative data, in many non-WEIRD countries, clinical neuroscientists often rely on using normative data from other populations. In South America, for example, tests validated and normed in Spain or Mexico are often used. This means that the normality of the performance of individuals is defined based on normative samples that are not all representative of the local populations. When such comparisons are analyzed in detail, the clinical hypothesis that is being tested can be revealed to be meaningless. Instead of the clinician using normative data to answer the question, ‘is this Peruvian individual’s task performance normal (for this individual)’, they find themselves actually asking, ‘is this Peruvian individual’s task performance normal for people who lived in Spain 20 years ago’.

Although the reasons for normative data being available only in the most-developed countries are complex, an unavoidable truth is that the appropriate use of normative data in cognitive assessments is only available for a minority of the human population. Continued use of normative data in WEIRD countries promotes a care gap between the WEIRD minority and the non-WEIRD majority. Methods of identifying cognitive impairments, other than based on norms and central tendency, that can be implemented in resource-limited contexts would help to reduce this imbalance. Some suggestions are given later in this paper.

## 8. Unfair Comparisons to Normative Data

Despite the lack of true representation in many normative samples, the intention is generally to compare the performance of individuals to their neurologically healthy peers. However, what should be considered appropriate peers is rarely considered. The standard practice is to generate normative tables that represent the entire population of the country (that is why representative samples are matched on census data). Thus, if an individual’s full IQ score is generated with the US version of the WAIS-IV, then their score indicates how close they are to the estimated population mean for the entire country. However, cognitive ability in practice varies by overlapping demographic factors, such as age, race, and socioeconomic status. Therefore, an individual could perform a cognitive test at a completely average level of performance for their demographic background, but their scores could be perceived as abnormally low compared to normative data that aims to represent the entire nation. This causes misdiagnoses and is unfair. Admittedly, several demographic adjustments are often attempted to rectify this, although often they fail to prevent normal diversity from being misidentified as an abnormality.

Age is an important factor, which is generally handled well by normative samples focused on central tendency. This is frequently achieved by specifying separate tables for different age groups. In fact, this is a necessity as the differential effect of age on different cognitive abilities would make them incomparable otherwise.

Race has also been treated in this way but, in contrast with age-adjusted norming, has caused more problems than it solved. The most controversial of these race-based normative tables is for the Halstead-Reitan Neuropsychological Test Battery for use in the USA [94]. These provide separate normative tables for Black and White individuals. While the aim of improving diagnostic accuracy with more precise tables is laudable, the consequences are that individuals will be more likely to be considered abnormal with one set of tables than the other. A recent legal dispute in the USA highlights the problems with using race-based norms. Players in the national football league (NFL) are at high risk of concussions and developing chronic traumatic encephalopathy [95], and many have sought compensation for cognitive impairments.

The dispute centers on the amount of compensation that is payable to injured players [96] based on their level of impairment as measured by the Halstead-Reitan battery. The more impaired players are, the greater their level of financial compensation. The problem has been that the central tendency of race-based norms was used to estimate each claimant’s premorbid ability (i.e., it was assumed that each player would have scored at the 50th percentile if they were cognitively unimpaired). The performance needed to be at the 50th percentile point is higher in the tables for white individuals than it is for black individuals. Therefore, given that cognitive impairment is being defined as how far current test performance is from the assumed premorbid 50th percentile, the estimated magnitude of impairment will be greater for any individual (black or white) if performance is evaluated on the White-derived normative tables than on the Black-derived normative tables. Consequently, given identical levels of cognitive test performance, white claimants were likely to receive more financial compensation for their neurological illness than black claimants. The neuropsychological and legal aspects of this case are described in detail by Gasquoine [27]. There is a growing consensus that the use of race-based norms in cognitive assessment should be scrapped.

Some have argued that race is an important concept within neurology that should be investigated [97]. In sharp contrast, the editors of the journal *Cortex* have strongly encouraged contributors to avoid the word ‘race’ entirely [98], citing evidence that race is not a concept that is scientifically tenable [99]. The editors reinforced this position in a later debate describing race as a pseudoscientific concept that cannot be used to add to knowledge simply because races do not exist [97]. However, they do point out that variation in socioeconomic status is an important factor in clinical neuroscience research.

Socioeconomic status is indeed an important demographic variable that predicts cognitive test performance [100,101], and part of that association is genetically mediated [102]. Therefore, if a clinician wishes to know how typical an individual’s test performance is, a fair comparison will involve comparing them to socioeconomically matched normative data. However, socioeconomic status is rarely considered a factor that should be accounted for when using normative tables. This has severe consequences for the over-pathologization of individuals who live in or grew up in below-average socioeconomic conditions.

At least in the USA, traumatic brain injuries are the second most common reason for referral for cognitive assessment in both pediatric [103] and adult settings [104] and thus comprise a large proportion of all clinical cognitive assessments performed. Traumatic brain injury patients are more likely to have a lower socioeconomic status [105] or live in poverty [106] than people who do not suffer such injuries. Nevertheless, patients with traumatic brain injuries will usually be evaluated against the normative sample, which is, by definition, of average socioeconomic status. This will inevitably lead to overestimation of the prevalence and severity of cognitive impairments. This observation extends beyond traumatic brain injury, as lower socioeconomic status is a risk factor for a wide range of neurological disorders in both pediatric and adult populations [107,108].

If we take, as a further example, an extreme of lower socioeconomic status, adults experiencing homelessness, several studies have shown that at the group level, they score about one standard deviation below the normative mean on cognitive tests [109,110]. This is often taken as evidence of cognitive impairment. However, to score one standard deviation below the national average is not at all unusual. Furthermore, people who experience homelessness very often come from relatively low socioeconomic status family backgrounds [111,112]. Therefore, a fair comparison when assessing for cognitive impairments would be of individuals from similar backgrounds. As this is rarely done, there are multiple published studies that ‘show’ high levels of cognitive impairments based on how well people experiencing homelessness have performed relative to the national average. One study described 64% of their USA-based sample of homeless youth as having cognitive ‘impairments’ based on them scoring more than one standard deviation below published normative mean scores [113]. Another reported study of language skills of women staying in a homeless shelter in the USA concluded that 60% of their sample had ‘deficits’ based on the scoring below thresholds anchored to normative central tendency points [114]. However, those rates of 60–64% would be exactly as expected if the samples were being incorrectly compared to a distribution centered on a mean for higher socioeconomic status individuals.

The lack of accounting for performance relative to similar peers, not simply the whole nation, has led to widespread over-pathologization of normal cognitive performance in people from lower socioeconomic status backgrounds, including homeless children and adults.

## 9. Conclusions and Tentative Suggestions to Achieve More Accurate and Fair Cognitive Assessments

I argue that the ideal level of performance in a cognitive assessment is, in practice, impossible to ascertain; following this, the consistent demarcation of non-ideal performance is also impossible. This may seem an academic matter, but it not for the importance of cognitive assessments in clinical, forensic, and educational decision making, where mismeasurement can have very serious negative consequences [115]. Historically, within clinical neurosciences that involve cognitive assessments, there has been a preference to identify ideal or typical task performance and, linearly from that, to define impaired or deficient performance. In this commentary, I have highlighted many of the problems with this approach. From a bio-evolutionary perspective, ideal performance points do not exist independently of environments. Furthermore, cognitive abilities may vary by type, not just ability, and individuals can use different cognitive strategies and degenerate neural systems to produce the behavior that the tests measure. Even if this were not the case, and typical (but not necessarily ideal) points could be identified, the psychometric distributions that are used to represent the test scores are, in practice, beset by artifacts and distortions. Furthermore, they are so often misapplied that they frequently over-diagnose cognitive impairments in the most vulnerable (e.g., the homeless) or, at times, underdiagnose cognitive impairments in other groups (e.g., black football players). They, therefore, have substantial potential for promoting injustice.

The practice within cognitive assessment of relying on central tendency to define normal and abnormal is also incompatible with the valuing of diversity. In the past, typicality has been erroneously equated with healthy such that normal diversity in abilities has frequently been identified as pathology. Many of the problems highlighted in this commentary, including compensation for sport-related neurological illness and over-pathologization of patients with traumatic brain injury or homelessness, stem from the implicit veneration of the average. However, this is not supported by biological analyses of variation in traits nor analysis of the degeneracy of normal neurocognitive functioning. So where now for assessment of abilities and traits, particularly cognitive assessment?

For neurodevelopmental conditions, there is already a movement underway: neurodiversity, which sets out a more accurate and fair way forward. The movement correctly recognizes that traits are either advantageous or disadvantageous, depending on the context [12,13,14,15]. It is argued that it may be best to focus on whether a particular extreme expression of a trait in an individual is associated with a disability rather than automatically defining it as indicative of a disorder [13]. However, it is also important to recognize that disability is not simply defined by ability but by its relationship to the environment [67]. As traits may be advantageous or disadvantageous based on the context, we need to be looking at which aspects of the environment are producing disability rather than which aspects of the individual are [15]. Thus, there is a need to consider the individual in the context in which they live rather than how they compare to others. This means adopting a more idiographic approach to understanding diversity within a developmental context.

For acquired cognitive disorders, there is the same need to move away from nomothetic analysis based on deviations from central tendency to an idiographic approach in which change within individuals is quantified. Many idiographic methods are already in use in cognitive assessment, and the call made here is simply to expand their application while reducing the use of nomothetic methods. Comparison to sample average scores is a nomothetic analysis, while premorbid estimation of function is idiographic [18]. Several methods already exist to estimate premorbid function, such as irregular-spelled word pronunciation, which can then be compared to observed current performance using existing normative tables and together provide a useful measure of post-injury ability [116]. Although these still require normative samples, they do not put any special value on the central tendency of performance, which is the core problem that is incompatible with valuing diversity. Employment of such methods provides a fairer method or evaluating changes in cognitive ability without introducing stigmatizing concepts about the abilities of different races or other factors [27].

Potential methods include estimation of premorbid function based on lexical reading, either as word pronunciation [117,118] or recognition [119,120], or estimation based on demographic variables [121] or scholastic records [122]. Two commonly used performance tasks in English that can be used for premorbid estimation of function are the word-reading task of the Wide Range Achievement Test and the Test of Premorbid Function [123]. For performance-based measures, co-norming of premorbid estimators with current performance measures allows for personalized, idiographic analysis of changes in cognitive ability. New tests are currently being developed and published, such as the Penn Reading Test, which is the premorbid estimation task included in the Penn Computerized Neurocognitive Battery [124]. That test is non-proprietary and can be accessed by anybody wishing to develop it further. Similarly, the English-language National Adult Reading Test [117] is open-access, as are equivalent versions of the test in many other languages. The commercially developed Test of Premorbid Function [118] was co-normed with the WAIS-IV and Wechsler Memory Scale-IV. Consequently, it is possible to estimate an individual’s premorbid IQ or memory ability based on either word pronunciation alone or in combination with demographic factors. Those scores can then be compared to the current IQ and memory performance without any need to compare them to ‘typically’ performing individuals. Furthermore, word pronunciation and other lexical reading tasks can be used to predict a range of cognitive functions [125], not just memory and IQ, and so could be extended to a wider range of cognitive tests.

Related to this, and already common practice within cognitive assessments, is to examine the jaggedness of cognitive profiles across multiple assessments [126] to detect possible changes at the level of the individual. This avoids the need to directly compare cases to what is considered ‘normal’. It has been argued, quite correctly, that cognitive assessment is more than psychometry, and the assessor must use their professional skill to examine the overall pattern of performance over multiple tests, including interpretation of the quality of errors, not just the quantity or accuracy [22].

A second approach is the development of regression-based norms to make personalized estimates of expected cognitive function [127]. It has recently been argued that multiple factors that impact brain health could be modeled, including socioeconomic status, educational quality, early life experiences, and racial discrimination, to provide highly personalized estimates of premorbid ability [96]. A particularly laudable and innovative approach was the recent publication of normative tables for neuropsychological tests for use with Guatemalan youth. This research used regression methods to adjust normative scores based on the individual’s vulnerability [128]. It, therefore, mitigates some of the problems described above that have led to overdiagnosis of disorders in the most vulnerable.

This approach, of personalized estimations, in contrast to nomothetic assumptions based on normative samples, has parallels with personalized medicine, which aims to tailor treatments to the individual patient [129]. Such methods still require aggregate statistics, and so some issues of the degeneracy of neurocognitive functions remain. Cognitive and brain scientists and psychometricians may be able to produce better assessment procedures that elucidate strategies used or constrain tasks such that only one brain system can effectively be applied. However, focusing on whether any drops in performance are associated with a disability may make some of those issues moot from a clinical care perspective.

A third method of refocusing assessments ideographically is to ask individuals or carers directly about what changes have occurred. Such an approach is already widely used in clinical assessments with the Frontal Systems Behavior Scale [130]. This scale quantifies changes in personality associated with neurological disease affecting the frontal lobes. Although neurological patients may often lack insight, informant systems of evaluating pre- and post-injury behaviors could be developed for other cognitive and behavioral issues, such as agitation, amnesia, and dysphasia. Some past research has already shown that these approaches may be useful for detecting the development of amnesia [131] and general cognitive decline [132]. Verbal report methods may also improve the identification of real-life problems rather than the abstract processes associated with most cognitive tests. For example, elderly people, particularly those with cognitive impairments, may have difficulty with financial planning and may be particularly susceptible to financial exploitation, likely reflecting executive dysfunction [133,134]. Such real-life executive function problems manifest within a wider context, which includes factors such as relatively lower education and premorbid ability (as estimated with word reading skill) [135] and the co-occurrence of depression [134]. Verbal reports allow a wider examination of contextual factors associated with real-life manifestations of cognitive impairments. Questionnaire-based assessments may also be substantially more reliable and valid than performance-based cognitive tests [136]. One example of the greater appreciation of intra-individual contextual factors is how cross-sectional studies of cognitive correlates of financial capacity [133] have benefited from longitudinal follow-up of patients [134].

Beyond individual cognitive assessments, in this commentary, I have also highlighted several studies that show that the inclusion of diverse study samples can bring new insights. Examples include the enhanced cognition seen in some children living in poverty around the world, and the benefits, not just costs, associated with carrying the APOE4 gene for children in Brazilian shanty towns.

In summary, I argue that clinical neuroscientists could benefit from increased awareness of the natural diversity of traits and abilities. In research, this could lead to new discoveries and more nuanced interpretations of known phenomena. In clinical cognitive assessments focusing on how individuals have changed rather than how average they are is fairer and avoids many of the problems inherent in the use of normative data. At the very least, there needs to be more careful application of normative data so that the performance of individuals is evaluated in the context of relevant peers.

When neuropsychologists, neuropsychiatrists, and behavioral neurologists recognize and value diversity over central tendency, decisions based on cognitive assessment can be made that are both more accurate and more just.

## Data Availability

No new data were created or analyzed in this study. Data sharing is not applicable to this article.

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
