# Peer review of "The Misguided Veneration of Averageness in Clinical Neuroscience: A Call to Value Diversity over Typicality"

_brainsci, 2023, doi:10.3390/brainsci13060860_

Round 1
Reviewer 1 Report
The manuscript is a commentary, more than a review of the literature, about the possible limit of comparing patients and controls, especially due to diffuse neurodiversity characteristics.
The manuscript is interesting and opens serious questions about the current approaches to neuropsychological evaluations and results, and could open an interesting discussion in the field.
My only concern is about the idea that this paper would be considered a review and not a commentary. I think it is not so balanced and unbiased to be considered as it is proposed by the author. For example, there is no mention of the strength of the current approach, and it seems that all the other researchers are publishing the wrong data. I think more cautions are needed. Or a more explicit statement that these are the author's ideas and that they should be used.
Author Response
Reviewer comment: The manuscript is a commentary, more than a review of the literature, about the possible limit of comparing patients and controls, especially due to diffuse neurodiversity characteristics.
Author response: I accept the reviewer’s point. The journal Brain Sciences has previously published articles classified as ‘Comment’ or ‘Opinion’, though these are not given as options in the Instructions for Authors (nor is ‘Commentary’. Perhaps the editor can decide whether the current manuscript would be better classified as something other than a review. As there is little that I can do about this as the author, I have at least edited the manuscript to not refer to it as a review, instead I have altered the text to self-refer to it as a commentary. These changes were made on lines 576, 591, and 690.
Reviewer comment: The manuscript is interesting and opens serious questions about the current approaches to neuropsychological evaluations and results, and could open an interesting discussion in the field. My only concern is about the idea that this paper would be considered a review and not a commentary. I think it is not so balanced and unbiased to be considered as it is proposed by the author. For example, there is no mention of the strength of the current approach, and it seems that all the other researchers are publishing the wrong data. I think more cautions are needed. Or a more explicit statement that these are the author's ideas and that they should be used.
Author response: The author accepts this point, and has made changes to clearly indicate that opinions are being expressed. There are changes to this effect in the abstract (lines 12 and 21). I have also added to text to the introduction section to emphasise that an opinion is being expressed, that new text is on lines 38-40. I have also made a small change in the final section that brings the ideas together to say that ‘I argue that’ (rather than stating as fact), on line 571. A similar change has been made in the penultimate paragraph (line 697). I feel that the manuscript now explicitly states that it contains the authors ideas and his belief that they should be applied.
Reviewer 2 Report
This is an opinion paper on an interesting issue. The author describes in enough detail the position of the paper. Some minor points that could help strengthen the quality of the paper is the use of real-life assessment (in neuropsychological research) examples to support the views of the author(s).
Regarding neuropsychological assessment on old age (strictly cognitive versus contextual with the use of questionnaires), you can see a discussion in:
Lichtenberg, P. A., Campbell, R., Hall, L., & Gross, E. Z. (2020). Context matters: Financial, psychological, and relationship insecurity around personal finance is associated with financial exploitation. The Gerontologist, 60(6), 1040-1049.
In addition to that in the discussion you could show the shift that researchers are already doing to the personalized (inter-individual) neuropsychological assessment perspective. This has been discussed in a relevant recent article regarding older adults:
Giannouli, V., Stamovlasis, D., & Tsolaki, M. (2022). Longitudinal study of depression on amnestic mild cognitive impairment and financial capacity. Clinical Gerontologist, 45(3), 708-714.
The above differs radically from the approach that the same researchers proposed with means and sds comparing different samples some years before (for the same topic):
Giannouli, V., Stamovlasis, D., & Tsolaki, M. (2018). Exploring the role of cognitive factors in a new instrument for elders’ financial capacity assessment. Journal of Alzheimer's Disease, 62(4), 1579-1594.
Author Response
Reviewer comment: This is an opinion paper on an interesting issue. The author describes in enough detail the position of the paper. Some minor points that could help strengthen the quality of the paper is the use of real-life assessment (in neuropsychological research) examples to support the views of the author(s).
Author response: New text has been added on lines 678-686 to introduce the ideas of real-life assessments and also the importance of context. This also required explanation of the Wide-Ranging Achievement Test, and so additional text about that was added on lines 626-629.
Reviewer comment: Regarding neuropsychological assessment on old age (strictly cognitive versus contextual with the use of questionnaires), you can see a discussion in:
Lichtenberg, P. A., Campbell, R., Hall, L., & Gross, E. Z. (2020). Context matters: Financial, psychological, and relationship insecurity around personal finance is associated with financial exploitation. The Gerontologist, 60(6), 1040-1049.
In addition to that in the discussion you could show the shift that researchers are already doing to the personalized (inter-individual) neuropsychological assessment perspective. This has been discussed in a relevant recent article regarding older adults:
Author response: New text has been added on lines 688-690 to emphasise that point. All three recommended citations have also been included in the revised manuscript.
Giannouli, V., Stamovlasis, D., & Tsolaki, M. (2022). Longitudinal study of depression on amnestic mild cognitive impairment and financial capacity. Clinical Gerontologist, 45(3), 708-714.
The above differs radically from the approach that the same researchers proposed with means and sds comparing different samples some years before (for the same topic):
Giannouli, V., Stamovlasis, D., & Tsolaki, M. (2018). Exploring the role of cognitive factors in a new instrument for elders’ financial capacity assessment. Journal of Alzheimer's Disease, 62(4), 1579-1594.
Round 2
Reviewer 1 Report
I think the paper can be accepted for publication due to its potential role in the debate in the neuropsychological field.